# Rethinking Reasoning in LLMs: Neuro-Symbolic Local RetoMaton Beyond ICL and CoT

**Rushitha Santhoshi Mamidala**                    SREERUSHITHA@USF.EDU

**Anshuman Chhabra**                               ANSHUMANC@USF.EDU

**Ankur Mali**                                     ANKURARJUNMALI@USF.EDU

*Bellini College of AI Cybsersecurity and Computing*

*University of South Florida, Tampa*

**Editors:** Leilani H. Gilpin, Eleonora Giunchiglia, Pascal Hitzler, and Emile van Krieken

## Abstract

Prompt-based reasoning strategies such as *Chain-of-Thought* (CoT) and *In-Context Learning* (ICL) have become widely used for eliciting reasoning capabilities in large language models (LLMs). However, these methods rely on fragile, implicit mechanisms often yielding inconsistent outputs across seeds, formats, or minor prompt variations making them fundamentally unreliable for tasks requiring stable, interpretable reasoning. In contrast, *automata-based neuro-symbolic frameworks* like **RetoMaton** offer a more structured and trustworthy alternative by grounding retrieval in symbolic memory with deterministic transitions. In this work, we extend RetoMaton by replacing its global datastore with a *local, task-adaptive Weighted Finite Automaton* (WFA), constructed directly from external domain corpora. This local automaton structure promotes *robust, context-aware retrieval* while preserving symbolic traceability and low inference overhead. Unlike prompting, which entangles context and memory in opaque ways, our approach leverages the explicit structure of WFAs to provide *verifiable and modular retrieval behavior*, making it better suited for domain transfer and interoperability. We evaluate this local RetoMaton variant on two pretrained LLMs **LLaMA-3.2-1B** and **Gemma-3-1B-PT** across three reasoning tasks: **TriviaQA** (reading comprehension), **GSM8K** (multi-step math), and **MMLU** (domain knowledge). Compared to the base model and prompting-based methods, augmenting these setups with local RetoMaton consistently improves performance while enabling transparent and reproducible retrieval dynamics. Our results highlight a promising shift toward *trustworthy, symbolic reasoning in modern LLMs* via lightweight, automaton-guided memory.

## 1. Introduction

Large Language Models (LLMs) have transformed Natural Language Processing (NLP) by demonstrating the ability to learn deep, generalizable knowledge from data (Petroni et al., 2019), enabling strong performance across tasks such as text translation, question answering, and human-like text generation (Sutskever et al., 2014; Ouyang et al., 2022; Qin et al., 2023; Chia et al., 2023). Although progress has been substantial, LLMs continue to face persistent challenges in mathematical reasoning and complex multi-step problem solving Dave et al. (2024), domains that demand structured and interpretable reasoning (Rae et al., 2021; Frieder et al., 2023). To bridge these gaps, techniques such as In-Context Learning (ICL) (Brown et al., 2020), and Chain-of-Thought (CoT) prompting (Wei et al., 2022) have been proposed to enhance reasoning and factual grounding without modifying model weights. However, each comes with inherent limitations: ICL, while effective,

demonstrates its strongest performance in large-scale models and is highly sensitive to the structure and ordering of prompts (Brown et al., 2020; Sclar et al., 2023; Razavi et al., 2025; Loya et al., 2023); and CoT prompting, though helpful for reasoning, can produce fragile outputs that hallucinate intermediate steps lacking logical consistency (Yeo et al., 2024). Meanwhile, task-specific fine-tuning of LLMs (Howard and Ruder, 2018) remains computationally intensive (Hanindhito et al., 2025; Yan et al., 2025), making it less practical for rapid adaptation. Moreover, having to fine-tune LLMs for specific reasoning problems detracts from their general-purpose nature and the knowledge encoded during the pre-training phase (Mou et al., 2016; Howard and Ruder, 2018). Thus, these limitations spanning computational overhead, latency, brittleness, and lack of interpretability underscore the ongoing challenges of achieving robust generalization and reliable reasoning in LLMs.

The persistent limitations of LLMs underscore the pressing need for structured and trustworthy mechanisms to elicit and ground reasoning processes in LLMs. One promising approach to address this challenge is through the integration of symbolic reasoning into neural models, a direction long pursued under the umbrella of Neuro-Symbolic AI (NeSy) (d'Avila Garcez et al., 2009; Besold et al., 2021). NeSy methods aim to combine inductive learning and generalization strengths of LLMs with the structured, interpretable inference offered by symbolic systems such as logic rules, automata and knowledge graphs (Manhaeve et al., 2018). This integration offers a principled way to overcome the opacity and fragility of purely neural models, enabling interpretable, modular, and context-sensitive reasoning. A key advantage of LLMs that often remains underutilized is their ability to encode text into rich, high-dimensional embedding spaces that capture the semantic and syntactic structure learned during pretraining (Mikolov et al., 2013; Devlin et al., 2019; Brown et al., 2020). This capacity, rooted in exposure to large and diverse corpora, enables generalization across tasks. While fine-tuning can further refine these representations for specific tasks, it is computationally expensive. We hypothesize that augmenting domain-specific knowledge directly in the embedding space can guide the model's behavior and improve generalization without the need for gradient updates. To realize this idea, we leverage RetoMaton (Alon et al., 2022), a neuro-symbolic extension of the kNN-LM framework (Khandelwal et al., 2019), that structures the embedding-based retrieval process using Weighted Finite Automata (WFAs). RetoMaton captures hidden representations from test corpora and organizes them into a symbolic structure that constrains retrieval during inference. This automaton-guided memory enables context-sensitive reasoning by enforcing structured and verifiable access paths, complementing the model's internal representations and offering an efficient, interpretable alternative to traditional fine-tuning.

In this work, we introduce the Local RetoMaton—a neuro-symbolic, task-specific datastore that integrates with LLMs via a WFA. Unlike global retrieval methods that sample from an entire corpus, Local RetoMaton builds its datastore from task-relevant text, ensuring every candidate aligns naturally with the target task. By constraining retrieval to this automaton-defined "local neighborhood," it reduces noise and enhances precision, selecting only the most pertinent contexts for each input. This tighter coupling between retrieved examples and the model's latent predictive manifold yields more accurate, better-calibrated predictions. As an unsupervised, nonparametric mechanism, Local RetoMaton persistently injects symbolic memory into the model in an architecture-agnostic manner without any fine-tuning or modification of the LLM itself. Consequently, it enables generalization beyond

the model's original training data while supporting structured, interpretable reasoning under uncertainty, a hallmark of neuro-symbolic systems (De Raedt et al., 2019; Garcez et al., 2019). Moreover, the Local RetoMaton complements prompting strategies by grounding them with structure knowledge enabling consistent, verifiable, and task-aware reasoning.

We evaluate the Local RetoMaton using the pretrained language models LLaMA-3.2-1B (Grattafiori et al., 2024) and Gemma-3-1B-PT (Team et al., 2025) on three distinct NLP tasks: reading comprehension (Zhu et al., 2021), mathematical problem solving (Ahn et al., 2024), and domain-general question answering (Yue, 2025). The symbolic component, WFA, is constructed from a task-specific datastore. For the TriviaQA dataset (Joshi et al., 2017), which targets reading comprehension, the WFA is built using associated evidence documents. For GSM8K (Cobbe et al., 2021) and MMLU (Hendrycks et al., 2021), which assess mathematical reasoning and general knowledge respectively, the WFAs are constructed using data from the training distribution of each dataset. Our empirical evaluations highlight that grounding LLMs through the Local RetoMaton framework yields several key benefits, including:

1. Improved **reasoning efficiency, enhanced generalization, and robust domain adaptation** achieved by injecting **non-parametric knowledge** in a structured manner using symbolic weighted finite automata (WFA).
2. **Improved consistency and robustness** across tasks by enforcing structured knowledge constraints.
3. The symbolic component allows **verifiable and interpretable** decision-making **enhancing transparency and explainability**.
4. Promotes **actionable and trustworthy** generation via **fine-grained** traversal.

Overall, using a task-specific Local RetoMaton improves LM's performance yielding an average gain of **4.48% with LLaMa and 2.78% with Gemma** over three downstream NLP tasks compared to the baseline model.

## 2. Related Works

We review two major directions in improving LLM reasoning: prompt-based generalization strategies and neuro-symbolic (NeSy) architectures that unify symbolic reasoning with neural representations. Together, these approaches aim to enhance interpretability, generalization, and trustworthiness in LLMs.

**Prompt-Based Generalization in LLMs.** While task-specific fine-tuning improves NLP performance (Howard and Ruder, 2018; Liu et al., 2019), it is computationally intensive (Radford et al., 2019; Ziegler et al., 1909) and lacks transferability. Prompting mitigates this by enabling inference without gradient updates. Evolving from manual templates to zero-, few-shot, and in-context learning (ICL) (Radford et al., 2019; Brown et al., 2020), ICL embeds task demonstrations directly into the input, guiding both format and behavior. Chain-of-Thought (CoT) prompting (Wei et al., 2022) extends ICL by including intermediate reasoning steps, which improves performance on arithmetic and multi-hop tasks by encouraging structured "thinking aloud." Complementary to prompting, retrieval-augmented approaches enhance generalization without retraining. kNN-LM (Khandelwal et al., 2019) interpolates predictions using nearest neighbors from an external datastore in the learned embedding space. RAG (Lewis et al., 2020) builds on this by retrieving raw text from ex-

ternal sources and injecting it into the prompt. While both approaches strengthen factual grounding, they suffer scalability limitations as datastore size grows.

**Neuro-Symbolic Approaches for Structured Reasoning.** To enhance explainability and compositionality, NeSy AI integrates logic-based reasoning with neural models (Garcez et al., 2019; d'Avila Garcez et al., 2009; Bhuyan et al., 2024). By embedding symbolic rules into differentiable systems, NeSy models combine the strengths of structure and generalization. Foundational efforts include Neural Theorem Provers (Rocktäschel and Riedel, 2017) and DeepProbLog (Manhaeve et al., 2018), which enabled differentiable reasoning over first-order logic and probabilistic rules. In language-based tasks, NeSy models have successfully mapped text to symbolic forms such as equations and expression trees (Roy and Roth, 2016; Chiang and Chen, 2018; Chen et al., 2019), allowing for structured, interpretable reasoning beyond shallow pattern matching. However, many such systems rely on supervised or semi-supervised data, limiting scalability.

RetoMaton (Alon et al., 2022) introduces a lightweight, architecture-agnostic NeSy framework inspired by kNN-LM. It structures the datastore as a WFA, clustering semantically similar embeddings into states and linking them with learned transitions. This enables efficient memory traversal across decoding steps by reducing redundant neighbor lookups while preserving context. Unlike other NeSy systems that depend on fine-tuning or external modules, RetoMaton integrates symbolic constraints directly into retrieval, enforcing coherent access paths and enabling structured generation without retraining.

## 3. Proposed Approach

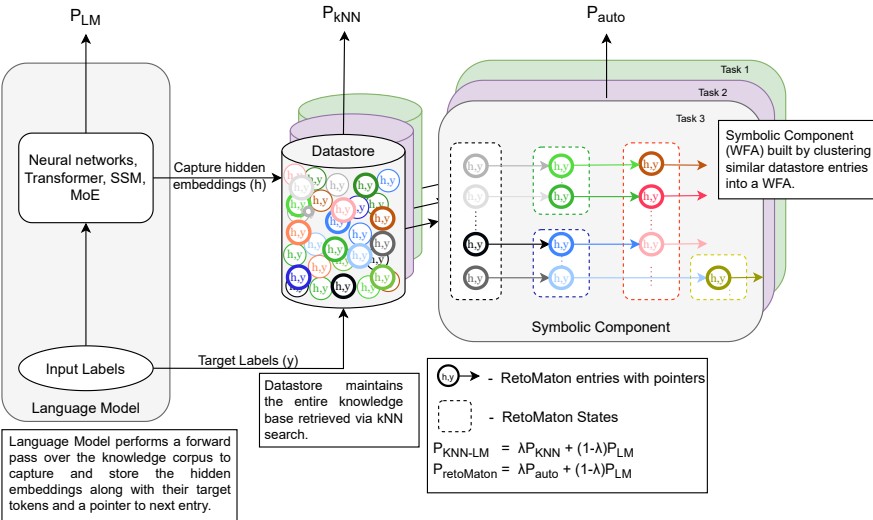

Figure 1: Overview of the Local RetoMaton framework. The system combines a language model, a symbolic datastore of hidden states and next-token labels, and a transition-structured automaton formed by clustering latent states.

We propose a theoretically grounded neuro-symbolic pipeline that transforms a frozen language model into an interpretable system using symbolic memory in the form of a Weighted Finite Automaton (WFA). Our framework is inspired by the RetoMaton design, but goes further by formalizing the integration of task-relevant representation space clustering with a more fine-grained local symbolic structure.

**From Language Model States to Symbolic Transitions:** Let $\phi\colon \mathbb{R}^d \times \Sigma \to \mathbb{R}^d$ denote the recurrence function of a pretrained language model, and let $[x_1, \ldots, x_n] \in \Sigma^n$ be a sequence from the corpus. The model produces a sequence of hidden states $h^t = \phi(h^{t-1}, x_t)$ for each time step $t$. Define the predictive target as $y^t = x_{t+1}$.

**Definition 1 (Transition Datastore)** *The symbolic datastore is a directed multigraph constructed from the sequence of hidden states and predicted tokens. Formally,*

$$D = \bigcup_{i=1}^{n-1} \{(h_i, y_i) \to (h_{i+1}, y_{i+1})\}, \tag{1}$$

*where each $h_i \in \mathbb{R}^d$ is the hidden state at position $i$, and $y_i = x_{i+1} \in \Sigma$ is the next token in the sequence. Each node in $D$ corresponds to a pair $(h_i, y_i)$, representing the hidden state and its associated predicted token label.*

**Definition 2 (State Abstraction via Clustering)** *Let $Q = \{q_1, \ldots, q_k\}$ denote clusters over $\{h_i\}$ learned using an unsupervised algorithm (e.g., k-means). Each cluster defines a symbolic state of a WFA.*

**Definition 3 (Weighted Finite Automaton with Representation Conditioning)** *We define the symbolic component as a WFA with vector-conditioned weights:*

$$(Q, \Sigma, q_0, \delta, \theta), \tag{2}$$

*where:*

- *$Q$ is a finite set of symbolic states,*
- *$\Sigma$ is the vocabulary,*
- *$q_0$ is the initial state,*
- *$\delta\colon Q \times \Sigma \to 2^Q$ is a non-deterministic transition function,*
- *$\theta\colon Q \times \mathbb{R}^d \times \Sigma \to \mathbb{R}_{\geq 0}$ assigns transition weights conditioned on hidden vectors.*

This transformation is **unsupervised**, **model-agnostic**, and requires no fine-tuning.

**Inference as Automaton-Guided Retrieval:** Let $h_q$ denote the current query vector and let $\mathcal{N}_k(h_q) \subset D$ be the $k$ nearest neighbors. Then retrieval operates over:

$$P_{\text{knn}}(y \mid h_q) \propto \sum_{(h_i, y_i) \in \mathcal{N}_k(h_q)} \mathbb{1}_{y=y_i} \cdot \exp\left(-\frac{\|h_q - h_i\|^2}{\mathcal{T}}\right). \tag{3}$$

The final token prediction is an interpolation with mixing coefficient $\lambda \in [0, 1]$,

$$P(y \mid h) = \lambda P_{\text{knn}}(y \mid h) + (1 - \lambda) P_{\text{LM}}(y \mid h), \tag{4}$$

After predicting token $y$, the datastore is filtered to successors with $y_i = y$, and pointers are used to advance to new hidden states $h_{i+1}$ forming the next candidate set $H_s$. Transitions are scored similar to Equation (3):

$$\theta(q, h, y) = \sum_{(h_i, y_i) \in s_q} \mathbb{1}_{y=y_i} \cdot \exp(-\text{dist}(h, h_i)), \tag{5}$$

$$P_{\text{ret}}(y \mid h) \propto \sum_{q \in s_q} \theta(q, h, y), \tag{6}$$

$$P(y \mid h) = \lambda P_{\text{ret}}(y \mid h) + (1 - \lambda)P_{\text{LM}}(y \mid h). \tag{7}$$

$s_q = \emptyset$, a global fallback kNN search is used.

**Symbolic Memory as Swappable External Knowledge:** The automaton-like structure enables interpretable, modular adaptation to new tasks. Datastores may be pruned, clustered differently, or constructed from task-specific corpora. This symbolic layer can be seen as a query-conditioned weighted automaton overlaying the language model's dynamics, enabling efficient memory, controllability, and symbolic introspection.

**Remark 4** *The clustering-based abstraction yields a finite state space $Q$, making the induced symbolic component strictly regular. Thus, the Local RetoMaton recognizes a regular language over the vocabulary $\Sigma$, grounded in the empirical transitions observed in the support corpus.*

**Efficiency Hypothesis:** By restricting retrieval to locally reachable transitions rather than the full datastore, the Local RetoMaton induces a bounded memory policy. This aligns with finite-state approximability and improves query-time complexity from $O(|D|)$ to $O(k+|s_q|)$, with controllable tradeoffs via $k$ and cluster granularity.

**Local vs. Global Retrieval Conjecture:** We conjecture that Local RetoMaton offers superior generalization and retrieval specificity over global retrieval mechanisms due to its structured symbolic memory. Readers are advised to look at Appendix C for detailed difference between global and proposed local RetoMaton.

**Conjecture 5 (Local Retrieval Generalization Hypothesis)** *Let $\mathcal{D}_{global}$ be a global datastore of unstructured $(h, y)$ pairs and let $\mathcal{D}_{local}$ be the same set organized into a finite-state automaton $\mathcal{A} = (Q, \Sigma, \delta, \theta)$ as in the Local RetoMaton. Then for a query embedding $h_q$ and target distribution $P(y \mid h_q)$, there exists a temperature $\mathcal{T}$ and mixing weight $\lambda$ such that:*

$$KL(P_{gold}(y \mid h_q) \| P_{local}(y \mid h_q)) < KL(P_{gold}(y \mid h_q) \| P_{global}(y \mid h_q)), \tag{8}$$

*where $P_{gold}$ is the true continuation distribution and $P_{local}, P_{global}$ are predictions from the local and global datastores, respectively.*

This hypothesis, supported by our empirical observation, reflects the assumption that locality-aware symbolic organization reduces retrieval noise and improves alignment with latent predictive structure. Empirically, this can be tested by measuring perplexity or KL-divergence on held-out continuations from task-specific corpora.

**Hyperparameters:** $k$ (retrieval size), $\lambda$ (interpolation weight), and $\mathcal{T}$ (temperature).

Thus we transform a support corpus into a structured symbolic memory aligned with an underlying language model, bridging connectionist and symbolic reasoning in a seamless, theoretically grounded way (Figure 1).

## 4. Experiments

The RetoMaton integrated NeSy LM is tested on three downstream NLP tasks: (1) Mathematical reasoning (2) General domain question answering (3) Reading Comprehension. In this section, our experiments demonstrate that we gain fine-grained insight into the NeSy LM's generation process rendering responses explainable, actionable, transparent,

and trustworthy; enabling domain adaptation and generalization; and ensuring reusability across tasks through a fail-safe WFA based RetoMaton.

**Datasets**

1. We evaluate mathematical reasoning using the **GSM8K** (Cobbe et al., 2021) dataset, which comprises 8.5K high-quality grade school math problems requiring multi-step reasoning with elementary arithmetic operations. Model performance is assessed on the *test* split using **accuracy** as the evaluation metric.

2. The **MMLU** (Hendrycks et al., 2021) benchmark includes 57 diverse tasks designed to assess both domain knowledge and problem-solving capabilities. We report performance on the official *test* set using **accuracy** as the primary evaluation metric.

3. The **TriviaQA** (Joshi et al., 2017) dataset evaluates reading comprehension by presenting question–answer pairs authored by trivia enthusiasts, each accompanied by independently gathered supporting evidence. This includes both `wiki_context` and `search_results` fields. We use the *validation* split for evaluation. Given the potential variability in phrasing, we report both **Exact Match (EM)** and **F1 score** to capture fully and partially correct responses.

**Experimental Setup** We primarily conducted our evaluations using 1B-parameter models: **LLaMA-3.2-1B** (Grattafiori et al., 2024) and **Gemma-3-1B** (Team et al., 2025). For each task, we used the best available snapshot of the model. Reading comprehension and general domain question answering were evaluated using **LLaMA-3.2-1B** and **Gemma-3-1B-PT**. For mathematical reasoning, we used the instruction-tuned versions **LLaMA-3.2-1B-Instruct** and **Gemma-3-1B-IT.** We employ a 5-shot ICL prompt for both the TriviaQA and MMLU datasets. For GSM8K, we use an 8-shot prompt with the LLaMA model and a 5-shot prompt with the Gemma model. All prompts are adapted from the publicly available LLaMA-Eval[1] benchmark suite to ensure consistency. For constructing the datastore and implementing the automaton-based retrieval infrastructure, we adapted code from Uri Alon's public implementation of RetoMaton[2], which uses FAISS (Johnson et al., 2019) for efficient similarity search. We experiment with hyperparameters for RetoMaton interpolation and retrieval using: $\lambda \in (0.1, 0.15, 0.2, 0.25)$, the number of nearest neighbors $k \in (1024, 512, 256)$, and temperature $\mathcal{T} \in (1, 0.95, 0.9, 0.85, 0.8)$, as detailed in Section 3. We conducted a grid search over these hyperparameters for each downstream task and report the best performance in our experiments. For decoding, we use a beam size of 5 and set maximum generation lengths to 10 tokens for MMLU, 175 tokens for GSM8K and 24 tokens for TriviaQA.

**Global RetoMaton** To explore the impact of grounding an LLM with external memory structured with a WFA, we constructed the RetoMaton from the WikiText (Merity et al., 2016) benchmark which we refer to as the Wiki Global RetoMaton. The WikiText dataset comprises well-curated, factually accurate, and broad domain Wikipedia articles that can support all downstream tasks. We evaluate the Global RetoMaton using a subset of the TriviaQA validation dataset and entire test splits of MMLU and GSM8K. The downstream performance of the LLaMA model augmented with the Global RetoMaton is presented in Figure 2.

---

1. Available on HuggingFace: https://huggingface.co/datasets/meta-llama/Llama-3.1-8B-evals https://huggingface.co/datasets/meta-llama/Llama-3.1-8B-Instruct-evals

2. https://github.com/neulab/knn-transformers

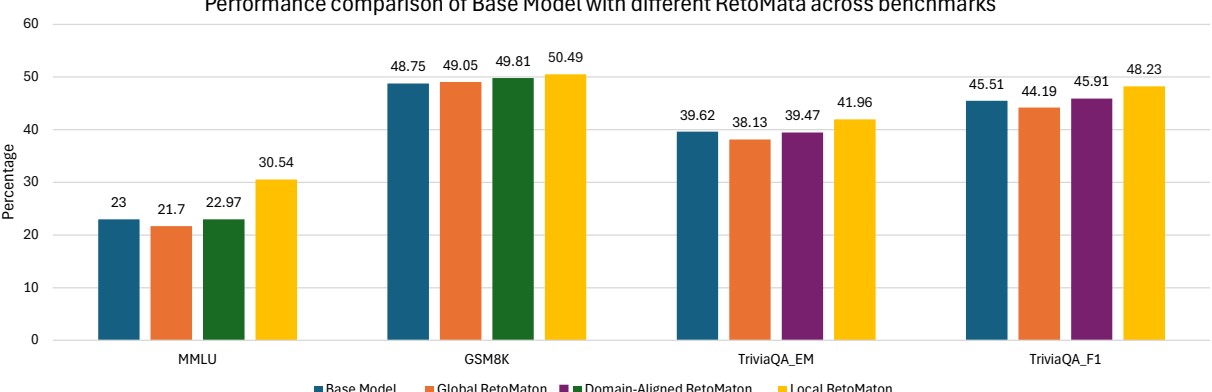

Figure 2: Comparison of downstream performance on MMLU (accuracy), GSM8K (accuracy) and TriviaQA (Exact Match & F1) for the baseline LLaMA model and its integrations with Global, Domain-Aligned, and Local RetoMata demonstrating that Local RetoMaton consistently delivers the highest performance.

Although integration of the Global RetoMaton shows no performance improvement across downstream tasks, using the symbolic component of the NeSy pipeline, we traced the nearest neighbors along the traversed paths and visualized the results in Figure 3. Additional traces provided in Appendix A.1 demonstrate generalization on GSM8K. This integration of external knowledge helped the model **generalize**, although it led to a slight drop in overall performance. Specifically, for GSM8K questions, 51% of decoding steps invoked new kNN searches and on TriviaQA this occurred in 64% of steps because when no valid paths remained, the model fell back on these searches as a **fail-safe** mechanism. By combining the RetoMaton's symbolic tracing with the LM, we achieved **explainable, transparent, and interpretable** responses, with WFA's path traversal providing truly **fine-grained** insights into the generation process. Additionally, once set up, the RetoMaton can be **repurposed across multiple tasks**, enabling efficient deployment. Building on these findings, we hypothesized that utilizing a corpus more closely aligned with the target domain would further enhance performance on downstream tasks.

Question 1: What was the name of Michael Jackson's autobiography written in 1988?

Output: Moonwalk (Correct Response)

Neighbors:
1. 1988, Jackson released his only autobiography," Moon"
2. 1988, Jackson released his only autobiography, Moon"walk"

Question 2: In which decade did stereo records first go on sale?

Output: 40s (Incorrect Response)

Neighbors:
1. too, or'30s and'"40"
2. style of the great soul ballads of the "60"

Figure 3: Text demonstrating retrieved RetoMaton entries from WikiText used for debugging, showing a TriviaQA question, ouput, and neighbors (enclosed in double quotes) along with their preceding context.

**Domain-Aligned RetoMaton** To support reading comprehension, we constructed a RetoMaton from TriviaQA's evidence documents; for mathematical reasoning on GSM8K and

the math domain of MMLU, we built a math-centric RetoMaton using the MathPile dataset (Wang et al., 2024) extracted by Shi et al. (2022) and Kim et al. (2024). The results in Figure 2 demonstrate that domain-aligned RetoMata outperform the Global RetoMaton across downstream tasks and, by revealing fine-grained insights into the generation process, guide **actionable** improvements. To push performance even further and explore whether a more tightly aligned distribution can surpass our baselines we now turn to the Local RetoMaton.

**Local RetoMaton** To draw a parallel with fine-tuning where model weights are updated on task-specific subsets we built datastores from the training splits of MMLU and GSM8K and, for TriviaQA, created a RetoMaton for each individual query. However, unlike fine-tuning, this process requires no parametric updates. While providing task-specific data, we constrain the scope to only the most relevant context, resulting in retomata that are significantly smaller in size when loaded into memory compared to a Global and Domain Specific RetoMata, making them more efficient to work with during inference. Retomaton information is provided in Appendix A.4.

Consistent with our hypothesis that locality-aware symbolic organization reduces retrieval noise and better aligns with the model's latent predictive structure, our empirical evaluation on k-shot examples shows a steady improvement from global to domain-aligned to local knowledge injection. Using the best-performing hyperparameters of the RetoMata on GSM8K, we measured Perplexity, KL divergence and negative log-likelihood by performing an evaluation pass and observed a clear, monotonic decrease across these strategies. As summarized in Table 1, the local retomata achieve the lowest values (PPL=2.7787; KLD = 0.0359; NLL = 1.0193), indicating that more task-localized symbolic datastores yield more precise, better-calibrated predictions. In Appendix C, we provide a formal analysis of Local RetoMaton's performance gains over Global RetoMaton and examine how clustering choices affect an LLM's performance.

Table 1: Comparison of perplexity (PPL), KL-Divergence (KLD), and negative log-likelihood (NLL) for the LLaMA model integrated with global, domain-aligned, and local datastores on GSM8K. The Local RetoMaton consistently achieves the lowest values across all metrics, indicating more accurate and better-aligned predictions.

|                          | PPL        | KLD        | NLL        |
| ------------------------ | ---------- | ---------- | ---------- |
| Global RetoMaton         | 4.0974     | 0.07466    | 1.3675     |
| Domain-Aligned RetoMaton | 3.6424     | 0.0534     | 1.2531     |
| Local RetoMaton          | **2.7787** | **0.0359** | **1.0193** |

**Cross-Model Evaluation** To assess the generalizability of our Local RetoMaton, we pair it with Gemma-3-1B language model and measure downstream task performance by integrating it with the Local RetoMaton. The results are demonstrated in Figure 4. The resulting gains on GSM8K and TriviaQA datasets mirror those observed previously, demonstrating that the improvements stem from the RetoMaton's symbolic component and are not tied to a specific model. Additionally, we have included the symbolic memory traces in Figure 6.

Figure 4: Downstream performance on MMLU (accuracy), GSM8K (accuracy) and Trivi-aQA (Exact Match & F1) for Gemma versus Local RetoMaton illustrating that the observed gains are attributable to the Local RetoMaton rather than model-specific effects.

## Discussion and Conclusion

We introduced **Local RetoMaton**, a neuro-symbolic augmentation mechanism that equips language models with *automaton-guided symbolic memory* enabling structured, interpretable, and context-sensitive reasoning. Unlike conventional prompting techniques, which rely solely on transient activations, RetoMaton integrates a persistent memory layer built as a weighted finite-state automaton over a local KNN-LM datastore. The creation process of this external memory is entirely unsupervised and does not require any parametric updates unlike finetuning. This memory structure provides *explicit control over retrieval paths*, allowing each inference step to be traced, understood, and manipulated. Our experiments across mathematical reasoning, question answering, and reading comprehension demonstrate that even compact models (e.g., 1B-parameter LLaMA and Gemma) benefit substantially from symbolic augmentation yielding both improved accuracy and introspectability. RetoMaton complements prompting strategies such as in-context learning (ICL) and chain-of-thought (CoT), offering a persistent memory backbone that grounds token-level predictions in task-aligned knowledge. Nonetheless, challenges persist in high-variance, heterogeneous settings like MMLU, where models often exhibit biased or default behavior. While RetoMaton imposes structure on retrieval, it cannot alone override biases embedded during pretraining. This suggests the need for adaptive symbolic scaffolding or hybrid corrective mechanisms to ensure faithful reasoning in open-domain tasks. Looking forward, we will investigate three key dimensions: (1) the effect of model scale on the integration of symbolic memory, where larger models may utilize structured retrieval more efficiently, (2) the generality of RetoMaton across diverse NLP tasks such as summarization, fact verification, and open-domain generation and (3) across diverse architectures like State Space Models and Mixture of Expert Models. We anticipate that symbolic augmentation will be particularly valuable for smaller or resource-efficient models, where external structure can compensate for limited internal abstraction. Ultimately, this work advances a concrete step toward **interpretable and controllable language models** grounded in the emerging paradigm of *Neuro-Symbolic AI* using the Local RetoMaton framework.

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

## Appendix A. Supplementary Retomaton Artifacts and Statistics

This appendix gathers all supplementary materials for our Retomaton experiments. First, we present the raw execution traces to illustrate workflow dynamics. Next, we list the exact prompts used to configure Local Retomata. We then report key statistics, namely token counts and space information for each setup. Finally, we formalize Global and Local retrieval strategies along with the impact of the cluster coefficient on LLM's performance.

### A.1. Wiki Global RetoMaton

Figure 5 illustrates several hidden representations traversed by LLaMA when integrated with the Global RetoMaton. By incorporating the Global RetoMaton's WikiText-derived knowledge, the model can leverage existing information to generalize more effectively to previously unseen inputs.

---

Input Question: Eliza's rate per hour for the first 40 hours she works each week is \$10. She also receives an overtime pay of 1.2 times her regular hourly rate. If Eliza worked for 45 hours this week, how much are her earnings for this week?

Output: Eliza's regular rate is \$10 per hour. For 40 hours, she earns 40 x 10 = 400 dollars. For 5 hours of overtime, she earns 5 x 10 x 1.2 = 60 dollars. 400 + 60 = 460. The final answer is 460 (Correct Response)

Neighbors indicated in double quotes along with their preceding context from WikiText:
1. who used the card to purchase one or more \$" "
2. workweeks averaging between 70 and 84 hours" per"
3. limit may be reduced ( 60 / " ="
4. 10 minutes will consume 40 x 6" x"

---

Figure 5: Text demonstrating retrieved RetoMaton entries from WikiText used for debugging, showing the GSM8K question, ouput, and neighbors (enclosed in double quotes) along with their preceding context.

### A.2. Symbolic Memory Trace

During generation, RetoMaton dynamically consults its symbolic memory. At each decoding step, RetoMaton retrieves representations from the WFA whose preceding hidden states are semantically close to the current context. The retrieved entries are then used to guide the next-token prediction. Figure 6 shows the top two datastore entries retrieved by Gemma model at each decoding timestep and color-coded with annotations provided for clarity.

Query: *What does the Latin phrase 'Fortes fortuna juvat' translate to in English?*
Answer: *Fortune favors the brave.*
**Timestep 1:**
    1. we read: Audentes Fortuna iuvat meaning : literally
                         previous context         current neighbor
    2. bold", while Fortes fortuna iuvat means literally Fortune
**Timestep 2:**
    1. been "fortes fortuna juvat" or fortune favors Fav
                                  valid predecessors
    2. English Audentes fortuna juvat in English  Fortune favors
**Timestep 3:**
    1. Audentes fortuna juvat in English Fortune favors the
    2. Webster Definition of fortes fortuna juvat :fortune favors the

Figure 6: Illustration of Symbolic Memory-Based Explainability: Retrieved neighbor tokens and their preceding contexts (annotated within) from the local datastore, shown during inference with the Gemma model integrated with the Local RetoMaton on the TriviaQA benchmark.

### A.3. Prompts

To setup the Local RetoMata for GSM8K and MMLU benchmarks, we captured hidden representations and next-token pairs from the training split, which consisted of zero-shot formatted examples, into an IVFPQ Faiss index. Figure 7 shows the input prompt structure used for constructing the Local RetoMaton for the MMLU benchmark with both models. Figure 8 shows the input prompt structures used to build the Local RetoMata for GSM8K using the LLaMa and Gemma models.

```
option = {0:'A', 1:'B', 2:'C', 3:'D'}
mmlu['option'] = option[mmlu['answer']]
mmlu['inputs'] = '''Answer the following multiple choice question. Choose the
correct answer by selecting the letter only (A, B, C, or D).
{mmlu['question']}
A. {mmlu['choices'][0]}
B. {mmlu['choices'][1]}
C. {mmlu['choices'][2]}
D. {mmlu['choices'][3]}
Answer: {mmlu['option']} '''
```

Figure 7: MMLU Input Format Used for Setting Up the Local RetoMaton

```
# LLaMa Model's input format
input = f'''<|start_header_id|>user<|end_header_id|>
Given the following problem, reason and give a final answer to the problem.
Problem: {example['question']}
Your response should end with \"\n#### [answer]\" where [answer] is the response
to the problem.<|eot_id|>
<|start_header_id|>assistant<|end_header_id|>

{example['answer']}'''

# Gemma Model's input format
input = f'''You are a helpful 2nd-grade math teacher. Help a 2nd grader to answer
problem in a short and clear manner. Your response should end with \"#### [NUM]\"
where [num] is the response to the problem.

Problem: {example['question']}

Answer: {example['answer']}'''
```

Figure 8: GSM8K Input Format Used for Setting Up the Local RetoMaton with both LLaMa and Gemma models

## A.4. Data Stores Statistics

The datastore details are summarized in Table 2, including the source text used for population, the number of tokens in each datastore, and the corresponding disk space occupied. Note that the TriviaQA datastore was built using only a 5,000 example subset of the dataset. The graph in Figure 9 shows the number of query-specific datastores constructed for TriviaQA, grouped by their respective size in megabytes (MB). While the 5k subset of TriviaQA resulted in a single datastore of size 9.8 GB, the query-specific RetoMata are significantly more lightweight, with each individual datastore being under 128 MB.

Table 2: Overview of RetoMata Datastores with Corresponding Token Counts and Disk Space

| Data corpus | # of Tokens | Size |
|---|---|---|
| Wikitext-103 | 121M | 8.13GB |
| MathPile | 187.2M | 12.57GB |
| TriviaQA | 146.9M | 9.87GB |
| MMLU | 38M | 2.57GB |
| GSM8K | 1.5M | 0.12GB |

Trivia QA's Query-Specific Datastores File Size Distribution

Figure 9: Distribution of file sizes for TriviaQA's query-specific datastores, showing that they are significantly more lightweight than global or domain-aligned indexes.

## Appendix B. Supplemental Results

### B.1. In-Context Learning Experiments

We conducted ICL experiments using the LLaMA model. We evaluated 3-shot, 5-shot, and 7-shot configurations using exemplar selection based on cosine similarity with the sentence-transformers/all-mpnet-base-v2 model(Song et al., 2020) adapting code from Gupta et al. (2023). Additionally, we included an 8-shot setup for GSM8K to match with RetoMaton experiments. ICL underperforms across tasks, in the case of MMLU largely due to domain mismatch in retrieved exemplars. This limitation highlights the value of our proposed approach that not only enhances performance but helps ground prompting startegies by providing a trustworthy and interpretable retrieval through structured symbolic memory. Table 3 reports the performance of the LLaMA model on downstream tasks using ICL, where the demonstration examples are selected based on cosine similarity computed over sentence embeddings.

Table 3: LLaMa Performance on Downstream Tasks: Optimal Prompts from the LLaMa Prompting Suite vs. k-Shot In-Context Learning

|  | TriviaQA | | MMLU | GSM8K |
|---|---|---|---|---|
|  | Exact Match | F1 | Accuracy | Accuracy |
| Local RetoMaton | **41.96** | **48.23** | **30.54** | **50.49** |
| LLaMa | 39.62 | 45.51 | 23 | 48.75 |
| 3-shot | 32.03 | 37.80 | 26.54 | 30.78 |
| 5-shot | 30.33 | 35.83 | 26.80 | 30.93 |
| 7-shot | 28.90 | 34.37 | 26.68 | 31.99 |
| 8-shot | - | - | - | 31.91 |

## Appendix C. Mathematical Theory: Global vs. Local RetoMaton Retrieval

### C.1. Setup and Definitions

Let $\Sigma$ be a finite alphabet. Let $D = \bigcup_{m=1}^{M} D^{(m)}$ be a dataset of $M$ sequences, each

$$D^{(m)} = \left\{ (h_1^{(m)}, y_1^{(m)}), \ldots, (h_{n_m}^{(m)}, y_{n_m}^{(m)}) \right\}$$

where $h_i^{(m)} \in \mathbb{R}^d$, $y_i^{(m)} \in \Sigma$.

Fix a clustering function $C : \mathbb{R}^d \to Q$ for some finite set $Q = \{q_1, \ldots, q_k\}$, and let $q_i^{(m)} := C(h_i^{(m)})$ be the cluster assignment for each hidden state.

**Key Design Choice:** All retrieval methods operate only on valid transitions, excluding sequence endpoints.

Define the empirical set of memory triples (valid transitions only):

$$\mathcal{D}_{\text{triple}} := \left\{ (h_i^{(m)}, y_i^{(m)}, q_{i+1}^{(m)}) : 1 \leq m \leq M, \ 1 \leq i < n_m \right\}$$

where $q_{i+1}^{(m)} := C(h_{i+1}^{(m)})$ is the cluster of the *next* hidden state.

For each $q \in Q$ and $y \in \Sigma$, define:

$$S(q) := \{ (h_i^{(m)}, y_i^{(m)}, q_{i+1}^{(m)}) \in \mathcal{D}_{\text{triple}} : q_i^{(m)} = q \}$$

$$S(q, y) := \{ (h_i^{(m)}, y_i^{(m)}, q_{i+1}^{(m)}) \in S(q) : y_i^{(m)} = y \}$$

Let $K : \mathbb{R}^d \times \mathbb{R}^d \to \mathbb{R}_{\geq 0}$ be a similarity kernel.

### C.2. Retrieval Probabilities

For any query $h \in \mathbb{R}^d$, let $q := C(h)$. Define the retrieval probabilities as:

**Global RetoMaton:**

$$P_{\text{global}}(y \mid h) := \begin{cases} \dfrac{\sum_{(h_i, y_i, q') \in \mathcal{D}_{\text{triple}}} \mathbb{1}_{y = y_i} K(h, h_i)}{\sum_{(h_i, y_i, q') \in \mathcal{D}_{\text{triple}}} K(h, h_i)} & \text{if denominator} > 0 \\ P_{\text{knn}}(y \mid h) & \text{otherwise} \end{cases}$$

where $P_{\text{knn}}(y \mid h)$ is standard $k$-nearest neighbor retrieval over all memories without kernel weighting.

**Local RetoMaton (Cluster-based):**

$$P_{\text{local}}^{\text{cluster}}(y \mid h) := \begin{cases} \dfrac{\sum_{(h_i, y_i, q') \in S(q)} \mathbb{1}_{y = y_i} K(h, h_i)}{\sum_{(h_i, y_i, q') \in S(q)} K(h, h_i)} & \text{if } S(q) \neq \emptyset \text{ and denominator} > 0 \\ P_{\text{global}}(y \mid h) & \text{otherwise} \end{cases}$$

**Local RetoMaton (Automaton-constrained):** For each token $y$ individually:

$$P_{\text{local}}^{\text{aut}}(y \mid h) := \begin{cases} \dfrac{\sum_{(h_i, y_i, q') \in S(q,y)} K(h, h_i)}{\sum_{(h_i, y_i, q') \in S(q)} K(h, h_i)} & \text{if } S(q, y) \neq \emptyset \text{ and } S(q) \neq \emptyset \\ P_{\text{local}}^{\text{cluster}}(y \mid h) & \text{if } S(q, y) = \emptyset \text{ but } S(q) \neq \emptyset \\ P_{\text{global}}(y \mid h) & \text{if } S(q) = \emptyset \end{cases}$$

**Note:** In automaton-constrained retrieval, if $S(q, y) = \emptyset$ (no empirical evidence for token $y$ from state $q$), we fall back to cluster-based retrieval for that specific token $y$.

### C.3. Main Lemma: Set Inclusion

**Lemma 6 (Support Set Inclusion)** *For all $q \in Q$ and $y \in \Sigma$:*

$$S(q, y) \subseteq S(q) \subseteq \mathcal{D}_{triple}$$

*Moreover, $\mathcal{D}_{triple} = \bigsqcup_{q \in Q} S(q)$ and $S(q) = \bigsqcup_{y \in \Sigma} S(q, y)$ are disjoint unions.*

**Proof** By definition, $S(q, y)$ consists of triples in $S(q)$ with the additional constraint $y_i^{(m)} = y$, so $S(q, y) \subseteq S(q)$. Similarly, $S(q)$ consists of triples in $\mathcal{D}_{\text{triple}}$ with $q_i^{(m)} = q$, so $S(q) \subseteq \mathcal{D}_{\text{triple}}$.

Every triple $(h_i^{(m)}, y_i^{(m)}, q_{i+1}^{(m)}) \in \mathcal{D}_{\text{triple}}$ has a unique cluster assignment $q_i^{(m)} = C(h_i^{(m)})$, so it belongs to exactly one $S(q)$. Similarly, within each $S(q)$, every triple has a unique token $y_i^{(m)}$, so it belongs to exactly one $S(q, y)$. ∎

### C.4. Main Theorem: Global as Special Case of Local

**Theorem 7 (Global-Local Equivalence for $k = 1$)** *If $|Q| = 1$ (i.e., $Q = \{q_*\}$), then for any $h \in \mathbb{R}^d$ and $y \in \Sigma$:*

$$P_{\text{global}}(y \mid h) = P_{\text{local}}^{\text{cluster}}(y \mid h) = P_{\text{local}}^{\text{aut}}(y \mid h)$$

**Proof** When $|Q| = 1$, every hidden state is assigned to the same cluster $q_*$, so $C(h) = q_*$ for all $h$. Therefore:

- $S(q_*) = \mathcal{D}_{\text{triple}}$ (all triples belong to the single cluster)

- $S(q_*, y) = \{(h_i, y_i, q') \in \mathcal{D}_{\text{triple}} : y_i = y\}$ for each $y$

Since $S(q_*, y) \neq \emptyset$ whenever token $y$ appears in the data, automaton-constrained retrieval never falls back. The denominators in all three cases are $\sum_{(h_i, y_i, q') \in \mathcal{D}_{\text{triple}}} K(h, h_i)$, and the numerators become identical for each $y$. ∎

### C.5. Corollary: Distinctness for Multiple Clusters

**Corollary 8 (Generic Distinctness for $k > 1$)** *Suppose $k > 1$ and the clustering function $C$ produces non-trivial clusters. Then there exist queries $h$ and tokens $y$ such that:*

1. $P_{\text{global}}(y \mid h) \neq P_{\text{local}}^{\text{cluster}}(y \mid h)$

2. $P_{\text{local}}^{\text{cluster}}(y \mid h) \neq P_{\text{local}}^{\text{aut}}(y \mid h)$

**Proof** [Sketch] If clusters are non-trivial, then for some $q$, we have $S(q) \subsetneq \mathcal{D}_{\text{triple}}$. When the kernel $K$ is non-degenerate (e.g., Gaussian), the restricted sum over $S(q)$ will generally differ from the full sum over $\mathcal{D}_{\text{triple}}$, establishing (1).

For (2), if some token $y \in \Sigma$ never appears from cluster $q$ in the training data, then $S(q, y) = \emptyset$. In this case, automaton-constrained retrieval falls back to cluster-based retrieval for token $y$, while for other tokens $y'$ with $S(q, y') \neq \emptyset$, it uses the restricted support. This creates different probability distributions. ∎

### C.6. Worked Example

Let $\Sigma = \{\mathtt{a}, \mathtt{b}\}$, $Q = \{q_1, q_2\}$.
  **Sequences:**

- Sequence 1: $(h_1^{(1)}, \mathtt{a}), (h_2^{(1)}, \mathtt{b}), (h_3^{(1)}, \mathtt{a})$

    – Cluster assignments: $C(h_1^{(1)}) = q_1, C(h_2^{(1)}) = q_2, C(h_3^{(1)}) = q_1$

- Sequence 2: $(h_1^{(2)}, \mathtt{b}), (h_2^{(2)}, \mathtt{a})$

    – Cluster assignments: $C(h_1^{(2)}) = q_1, C(h_2^{(2)}) = q_2$

  **Memory triples in $\mathcal{D}_{\textbf{triple}}$:**

- From sequence 1: $(h_1^{(1)}, \mathtt{a}, q_2), (h_2^{(1)}, \mathtt{b}, q_1)$

- From sequence 2: $(h_1^{(2)}, \mathtt{b}, q_2)$

  **Support sets:**

- $S(q_1) = \{(h_1^{(1)}, \mathtt{a}, q_2), (h_1^{(2)}, \mathtt{b}, q_2)\}$

- $S(q_2) = \{(h_2^{(1)}, \mathtt{b}, q_1)\}$

- $S(q_1, \mathtt{a}) = \{(h_1^{(1)}, \mathtt{a}, q_2)\}$

- $S(q_1, \mathtt{b}) = \{(h_1^{(2)}, \mathtt{b}, q_2)\}$

- $S(q_2, \mathtt{a}) = \emptyset$ (token $\mathtt{a}$ never observed from state $q_2$)

- $S(q_2, \mathtt{b}) = \{(h_2^{(1)}, \mathtt{b}, q_1)\}$

**For a query $h$ with $C(h) = q_2$:**

- $P^{\text{aut}}_{\text{local}}(\texttt{a} \mid h) = P^{\text{cluster}}_{\text{local}}(\texttt{a} \mid h)$ (fallback, since $S(q_2, \texttt{a}) = \emptyset$)

- $P^{\text{aut}}_{\text{local}}(\texttt{b} \mid h) = \frac{K(h, h_2^{(1)})}{K(h, h_2^{(1)})} = 1$ (only observed token from $q_2$)

This shows how automaton constraints can eliminate certain predictions while falling back gracefully for unobserved transitions.

### C.7. Theoretical Remarks

1. **Consistent Transition View:** All methods operate on the same space of valid transitions $\mathcal{D}_{\text{triple}}$, ensuring fair comparison.

2. **Graceful Degradation:** The fallback hierarchy (automaton $\rightarrow$ cluster $\rightarrow$ global) ensures robust probability estimates even with sparse data.

3. **Empirical Constraint:** Automaton-constrained retrieval respects empirical evidence— if a transition $(q, y)$ was never observed, it defers to less restrictive methods.

4. **Ultimate Fallback:** The complete fallback hierarchy is automaton $\rightarrow$ cluster $\rightarrow$ global $\rightarrow$ $k$-NN, ensuring robust probability estimates under all conditions including kernel failure.

