# OpenReview forum: "Rethinking Reasoning in LLMs: Neuro-Symbolic Local RetoMaton Beyond CoT and ICL"
_nesyconf.org/NeSy/2025/Conference_Phase_2 — NeSy 2025 - Phase 2 Poster_

### Official Review · Reviewer_Qmmz · 2025-06-23
**Local RetoMaton: Promising Direction, But Lacking Experimental Clarity**

**Rating:** 5
**Confidence:** 3

**Review:**

Revised Version:
This paper presents Local RetoMaton, a neuro-symbolic extension to Large Language Models (LLMs) that integrates a task-aligned Weighted Finite Automaton (WFA) to augment memory retrieval during inference. The approach aims to address the fragility and opacity of prompt-based reasoning strategies such as In-Context Learning (ICL) and Chain-of-Thought (CoT) prompting. The authors evaluate their method on three benchmarks, TriviaQA, GSM8K, and MMLU, using small-scale LLaMA and Gemma models. Results demonstrate modest improvements in accuracy, along with the added benefit of interpretable and structured reasoning paths.

Strengths:
- The paper targets a limitation of current LLM reasoning methods, namely, their lack of interpretability.
- Local RetoMaton offers a lightweight augmentation, requiring no fine-tuning or additional model parameters.
- The empirical evaluation spans three diverse reasoning tasks.

Weaknesses:
1. Insufficient Differentiation Between RetoMaton and Local RetoMaton:
   The manuscript does not clearly articulate how Local RetoMaton significantly extends or differs from the original RetoMaton framework. While the introduction of locality and task-aligned datastores is implied to be novel, these aspects should be further discussed. Providing concrete examples of these datastores would enhance clarity.

2. Limited Experimental Depth and Visualization:
   The experimental section would benefit from clearer, more systematic presentation of results, ideally in tabular form. Current figures are somewhat cluttered. Furthermore, claims regarding robustness and interpretability remain largely qualitative and should be supported with quantitative evidence.

3. Missing Ablation Studies:
   The paper would be significantly strengthened by ablation experiments that examine:
   - The contribution of local versus global retrieval
   - The impact of WFA clustering granularity
   - The effect of temperature and interpolation hyperparameters

4. Lack of Quantitative Interpretability Metrics:
   While symbolic memory traces are provided, no metric-based evaluation of interpretability or consistency is included. For example, measuring retrieval stability under prompt perturbations or computing KL divergence between retrieval paths could help substantiate interpretability claims.

While Local RetoMaton introduces a promising and potentially impactful idea, the current manuscript lacks the clarity, experimental rigor, and analytical depth needed to fully validate its contributions. A clearer distinction from prior work, particularly the original RetoMaton, and a more comprehensive evaluation are recommended.

**Anonymity:**

Remain anonymous

---

### Official Review · Reviewer_juMP · 2025-07-07
**Interesting and throroughly evaluated extension of RetoMaton.**

**Rating:** 7
**Confidence:** 3

**Review:**

Abstract: Is well written

Novelty: Is clear but limited, due to slight modification of RetoMaton approach.

Text in general: Is well written clear and understandable. The text could guide readers, who are not experts in the field a bit better, through the paper.

Clarity and introduction of variables: Due to some variables being introduced late or not at all, the clarity is limited and the paper is hard to read at times. The text should introduce all used variables in a full sentence instead of in a key word fashion, and exactly before first or directly after first occurence. Mathematical notation like orthogonal signs that even include a strike through should be introduced and not considered to be known to the reader. Furthermore, not all equations are enumerated, but should be.

Overall presentation and figures: Figure 1 can benefit from some high-level explanatory add-ons. Figure 1, 2, and 3 are presented in too small font sizes and lack a main message as a caption. The captions state descriptions of the obvious instead. Also x axis labels with under-bars in their names should be changes to more appropriate names.

Numerical Results: The accuracies between the baseline models and the novel contribution do not deviate a lot, but show clearly that the method outperforms the baseline. I am not sure, if there are besides testing against the baseline alternative comparative approaches to compare, if so this should be done.

**Anonymity:**

Remain anonymous

---

### Official Review · Reviewer_oGik · 2025-07-08
**The paper’s contributions to neuro-symbolic reasoning, strong empirical results, and alignment with NeSy 2025 are acceptable. However, the authors should address the following points.**

**Rating:** 8
**Confidence:** 4

**Review:**

This paper uses Local RetoMaton that extends the original RetoMaton by introducing task-specific WFAs, offering a lightweight, model-agnostic alternative to fine-tuning or prompting. This is a significant step toward scalable neuro-symbolic reasoning, especially for resource-constrained settings.

1. The experimental setup (Page 7) mentions adapting prompts from the LLAMA-Eval benchmark suite and using 5-shot ICL, but details on prompt design, especially for CoT baselines are sparse.

2. Additionally, the hyperparameter grid search (k,  λ, T) is described, but the impact of individual parameters on performance is not analyzed.

3. Constructing task-specific WFAs for each query in TriviaQA may be computationally intensive, especially for large datasets. The paper does not quantify the preprocessing cost or memory overhead of building these WFAs.

**Anonymity:**

Remain anonymous